behaviour, biomechanics, ecology

adaptation, *Bufo marinus*, jumping kinematics, dispersal, invasive species

**Author for correspondence:**
Richard Shine
e-mail: rick.shine@mq.edu.au

# The accelerating anuran: evolution of locomotor performance in cane toads (*Rhinella marina*, Bufonidae) at an invasion front

Cameron M. Hudson[1,2], Marta Vidal-García[3,4], Trevor G. Murray[3,5] and Richard Shine[1,5]

[1]School of Life and Environmental Sciences, The University of Sydney, New South Wales 2006, Australia
[2]Department of Fish Ecology and Evolution, Eawag, Swiss Federal Institute of Aquatic Science and Technology, Centre of Ecology, Evolution and Biochemistry, Seestrasse 79, 6047 Kastanienbaum, Switzerland
[3]Research School of Biology, The Australian National University, Canberra, Australian Capital Territory 2601, Australia
[4]Department of Cell Biology and Anatomy, University of Calgary, 3330 Hospital Drive NW, Calgary, Alberta, Canada
[5]Department of Biological Sciences, Macquarie University, New South Wales 2109, Australia

(iD) CMH, 0000-0003-3298-4510; MV-G, 0000-0001-7617-7329; RS, 0000-0001-7529-5657

As is common in biological invasions, the rate at which cane toads (*Rhinella marina*) have spread across tropical Australia has accelerated through time. Individuals at the invasion front travel further than range-core conspecifics and exhibit distinctive morphologies that may facilitate rapid dispersal. However, the links between these morphological changes and locomotor performance have not been clearly documented. We used raceway trials and high-speed videography to document locomotor traits (e.g. hop distances, heights, velocities, and angles of take-off and landing) of toads from range-core and invasion-front populations. Locomotor performance varied geographically, and this variation in performance was linked to morphological features that have evolved during the toads' Australian invasion. Geographical variation in morphology and locomotor ability was evident not only in wild-caught animals, but also in individuals that had been raised under standardized conditions in captivity. Our data thus support the hypothesis that the cane toad's invasion across Australia has generated rapid evolutionary shifts in dispersal-relevant performance traits, and that these differences in performance are linked to concurrent shifts in morphological traits.

## 1. Introduction

To understand the evolutionary processes that have shaped phenotypic variation within and among species, we ideally need information not only on morphology, but also on the ways in which morphological variation translates into effects on performance, and hence on lifetime reproductive success [1–3]. For many traits, such evidence is difficult to gather. For example, a trait may be invariant within a broad phylogenetic lineage (e.g. viviparity in mammals) or may have complex and difficult-to-document effects on a range of performance measures (e.g. brain morphology versus cognitive ability). Also, the evidence may be weakened by the need to rely on comparisons between taxonomic entities that have been separated for long periods of time and hence may have accumulated many differences unrelated to the trait of interest. For these reasons, some of the best opportunities to explore adaptive evolution come

from systems in which change has occurred very rapidly and involves characteristics whose performance attributes are readily measurable.

Locomotor ability in invasive species satisfies these criteria. Individuals at the expanding edge of an invasion often exhibit enhanced rates of dispersal relative to conspecifics from range-core populations [4], and at least some of those cases reflect heritable changes rather than phenotypic plasticity [5,6]. Variation in locomotor performance is easier to quantify than many other biological functions, and that variation often has an intuitive link to fitness. For example, greater speed may enhance survival because a faster individual can capture prey or escape predators more effectively [7–11]. Likewise, for dispersal ability, individuals that move further may have access to new resources and/or experience reduced competition with conspecifics or kin [12–15].

Despite the potential management implications of enhanced dispersal ability at the expanding edge of invasions, most analyses of this topic have not directly measured locomotor ability. Instead, they have documented morphological changes, and inferred that those changes enhance an individual's ability to disperse [16–18]. Although in some cases those inferences are strong, ideally we need empirical evidence not only about morphological changes during an invasion, but also on changes in locomotor ability, and the relationship between those two parameters.

One of the most intensively studied cases of accelerated spread involves the invasion of the cane toad (*Rhinella marina*) through tropical Australia [19]. Extensive data are available to document the increasing rate of spread overall [20] and to link that acceleration to faster dispersal of individuals at the invasion front compared with the range core [21,22]. Invasion-front toads also exhibit distinctive morphologies [23,24], physiologies [25–27] and behaviours [28–30] that have evolved rapidly during the colonization of Australia. By raising individuals from different populations in a common-garden experiment, researchers have demonstrated that some of these dispersal-enhancing traits are heritable [5,24,25,29,30]. Evolved shifts in morphology have been interpreted as adaptive responses to the benefits of faster dispersal [13,24,31,32], but the critical links between variation in morphology and locomotor performance have attracted little attention. In the field, toads with relatively longer legs dispersed faster than their short-limbed conspecifics [31]— but the relationships (if any) between morphological traits and performance traits remain largely unstudied.

To fill this gap in knowledge, we examined the morphology and locomotor performance (and relationships between those parameters) of cane toads from range-core and invasion-front populations. We predicted that range-core and invasion-front toads would differ both in morphology and performance, that these differences would be seen in captive-raised (common-garden experiment) progeny as well as in wild-caught animals, and that morphological and locomotor variables would be significantly correlated.

## 2. Material and methods

### (a) Study species
Native to a large range in South America, cane toads were brought to Australia (via Puerto Rico and Hawai'i) in 1935, as a control for insect pests of commercial sugar-cane farming [19]. Thousands of the progeny of the introduced toads were released in coastal areas of Queensland (QLD). Over the next 85 years, the toads spread westwards across QLD, the Northern Territory and Western Australia (WA) at an accelerating pace [20]. Toads at the western invasion front differ morphologically, physiologically and behaviourally from those in the eastern range core, and many of those differences are heritable (see Introduction and references cited therein).

### (b) Study animals and jumping performance trials
Adult toads (snout–vent length (SVL) > 90 mm) were collected from four invasion-front populations (El Questro, Oombulgurri, Purnululu, Wyndham; all 0–3 years post-colonization) in WA and three long-colonized populations (Innisfail, Townsville, Tully; all 80 years post-colonization) in QLD from October to December in 2013. These individuals were maintained in captivity at the University of Sydney Tropical Ecology Research Facility (Northern Territory: 12°37′ S, 131°18′ E) as part of a 'common-garden' breeding experiment (see [32] for details). In October 2014, we tested locomotor performance of 195 wild-caught toads (F0) by filming them as they hopped over a 5 m$^2$ area of hard bare ground (soil). Between 19.00 and 00.00 h, toads were filmed at 240 fps using two high-speed (HS) filming cameras (model Casio Exilim EX-ZR1000), which were placed on tripods (10 cm off of the ground, 20 cm apart) forming an angle between them of 140°, along with portable work lights (model IronHorse 24 LED), a camera flash and a calibration checkerboard. All individuals filmed were encouraged to jump by gently tapping them on the urostyle with a blunt pole, a common procedure for inducing jumping in anuran amphibians [33]. After each filming trial, we used a camera flash in order to synchronize the two high-speed filming cameras (left and right), and we placed a checkerboard in several locations and orientations within the jumping track in order to calibrate the cameras and reconstruct the dimensions and positions of the three-dimensional (3D) space for each trial. Morphological characteristics (mass, SVL, head width, lengths of the forefeet, hindfeet, radioulna, humerus, femur and tibiofibula) were measured (by C.M.H.) with an electronic balance and Vernier calipers (to the nearest 0.1 mm) for each individual following trials. We ran a second set of trials in March 2016 using captive-raised (F1) individuals (progeny from the common-garden experiment, 19–24 months of age at the time of testing), following the same procedure as above. Ambient temperatures during the trials ranged from 24.9 to 28.5°C.

### (c) Raceway trials
In a separate set of trials, toads were placed by hand at the start of a 2 m wide, 15 m long outdoor raceway and encouraged to hop by prodding their urostyle with a blunt pole. This raceway was constructed on hard bare ground (soil) and had 0.5 m high walls to prevent escape. Trials commenced immediately after the toad was placed on the racetrack. Individuals that refused to hop after 10 consecutive pokes were considered to be exhausted or unwilling to move, and their trial was terminated. We also recorded the time and number of hops for the toad to complete each 5 m segment of the racetrack. All trials were conducted between 19.00 and 00.00 h, at ambient temperatures of 25.4–27.4°C.

### (d) Video analyses
High-speed videos from both views (left and right cameras) were split into frames using QuickTime Pro v. 7.7.4. These frame sequences were loaded into the Digilite Toolbox v. 1.4 (ANU Visual Sciences Group) of MATLAB v. 8.2.0.701 [34]. The entire jump sequence (take-off to landing) was scored by placing digital landmarks on the snout and cloaca of a toad for each frame, resulting in a series of points that define a discrete jump. We

calibrated each of our filming trials using the Camera Calibration Toolbox for MATLAB [35], which uses an optimized implementation to rectify frame sequences from both cameras (left and right) and creates a 3D space framework that can integrate the digitized coordinates from both views (left and right). We selected 10–20 synchronized pairs of images from the left and right videos containing the calibrating planar checkerboard and processed them with the Camera Calibration Toolbox for MATLAB [35], using the two-dimensional (2D) calibration and stereo-calibration (3D) functions. We controlled the accuracy of all calibrations and stereo-calibrations by suppressing any images that displayed more than 1% inaccuracies between the left and right views.

We obtained time-stamped, real-world 3D trajectories of the snout and cloaca by processing the digitized landmark information through this calibrated 3D space framework using batch processor for the MATLAB Toolbox Synchronise [36]. Because the 3D coordinates had been gathered from different trials, the position of the $xz$-coordinate plane differed across specimens. In order to make 3D jumping trajectories comparable across all specimens, we standardized the position of all $xz$-coordinate planes by rigidly rotating each 3D coordinate to the $(x, 0, z)$ plane, following the methodology from Vidal-García et al. [37] and using internal functions from the R package ShapeRotator [38]. With these aligned 3D coordinates (in cm) of the snout and cloaca for each individual, and in each frame, we extracted the following kinematic variables: mean velocity (cm s$^{-1}$), maximum velocity (cm s$^{-1}$), distance (cm), height (cm) and jumping angles (calculated between the $xz$-plane and the vector from the cloaca to the snout) at take-off and at landing. Code for obtaining these kinematic variables is provided in our GitHub repository (https://github.com/marta-vidalgarcia/jumping-toads) and the Dryad Digital Repository.

## (e) Comparisons between jumping performance and raceway trials

Our earlier analysis of raceway performance in 449 Australian cane toads revealed differences between range-core and invasion-front individuals in behavioural responses; notably, invasion-front toads were reluctant to run and hence were slower [32]. Some of these animals ($n = 89$) were the same as those for which we measured jumping performance (above). We re-analysed the raw data from the raceway study to ask if distances covered per hop were significantly linked to the morphology of toads, and whether an individual's performance on the raceway was correlated with its jumping performance in the video trials.

## (f) Statistical analyses

To remove effects of body size on other dimensions, we regressed each morphological variable (e.g. femur length) against the animal's body length (SVL), and we used the residual scores from those regressions as our measures of body shape. Analyses of locomotor performance (from jumping and raceway trials) were based on a single value per individual; in cases where we had data for two jumps per animal, we used data from the longest hop. Because of strong ontogenetic and sexual differences in body sizes and body proportions [24], we conducted some analyses separately for males, females and juveniles (individuals less than 90 mm SVL, and thus too small for us to determine sex). Data for wild-caught and captive-raised individuals also were analysed separately to explore some issues, and combined for others.

Initial MANOVAs on morphology, with invasion category (range core versus invasion front) and population nested within invasion category as factors, and morphological traits as the repeated measure, showed that morphological traits differed between invasion categories in different ways (interaction between invasion category and morphological traits for all toads $F_{7,294} = 3.90$, $p < 0.001$; for F0 only, $F_{7,159} = 7.22$, $p < 0.0001$; for F1 only $F_{7,120} = 3.43$, $p < 0.003$). The same was true for our locomotor measures (interaction between invasion category and jumping performance variables for all toads $F_{5,289} = 2.87$, $p < 0.02$; for F0 only, $F_{5,156} = 2.42$, $p < 0.04$; for F1 only $F_{5,120} = 3.94$, $p < 0.003$) so we proceeded to explore geographical variation in each trait separately. To clarify these patterns, we used ANOVA with invasion category (range core versus invasion front) and population nested within invasion category as factors, and either morphological or performance measures as dependent variables. The contribution of morphology to the variation in performance was assessed through multiple regression analyses (backward stepwise elimination). To quantify links between traits such as alternative measures of performance, or morphology versus performance, or between data from jumping versus raceway trials, we calculated standard (Pearson's product-moment) correlation coefficients. JMP v. 14.0 (SAS Institute, Cary, NC, USA) was used for all statistical analyses.

## 3. Results

### (a) Sample sizes and body sizes
We obtained data for 311 toads (138 females, 128 males, 45 juveniles) from three range-core populations (Innisfail $n = 46$, Townsville $n = 49$, Tully $n = 53$) in QLD and four invasion-front populations (El Questro $n = 44$, Oombulgurri $n = 35$, Purnululu $n = 32$, Wyndham $n = 52$) in WA. The mean body masses averaged larger in females than in males, and larger in adults than in juveniles, but were similar within each of these groups in WA and QLD (ANOVA main effect of sex/age class $F_{2,305} = 49.31$, $p < 0.0001$; state $F_{1,305} = 2.71$, $p = 0.10$; interaction sex × state $F_{2,305} = 0.21$, $p = 0.81$; tables 1 and 2). Of the 235 toads in total, 174 were wild-caught and the other 137 were captive-raised progeny.

### (b) Effect of location on body shape
ANOVAs with invasion category (range core versus invasion front) and population nested within invasion category as factors, and residual scores as dependent variables, showed that toads from the two extremes of the species's Australian distribution differed in many traits (figure 1 and tables 1 and 2). First, we look at data from wild-caught adult animals. For female toads, range-core and invasion-front individuals differed significantly in the relative length of the limb bones (femur, radioulna) and forefeet, and width of the head, with invasion-front females larger in each case. For male toads, these geographical differences were significant for length of the tibiofibula (range core larger; figure 1 and table 2).

If we restrict analysis to captive-raised progeny only (i.e. excluding wild-caught animals), at least one sex/age class showed statistically significant differences between invasion-front and range-core populations for all traits (adult females for head width and tibiofibula length; adult males for relative forefoot length; juveniles for all traits: table 2).

### (c) Variation in jumping performance
The 311 hops that we analysed spanned a wide range in most of the attributes that we measured. For example, the mean velocity ranged from 95.1 to 389.5 cm s$^{-1}$, the distance covered from 22.8 to 119.9 cm and the maximum height aboveground from 2.2 to 23.9 cm. Many of the variables that we measured

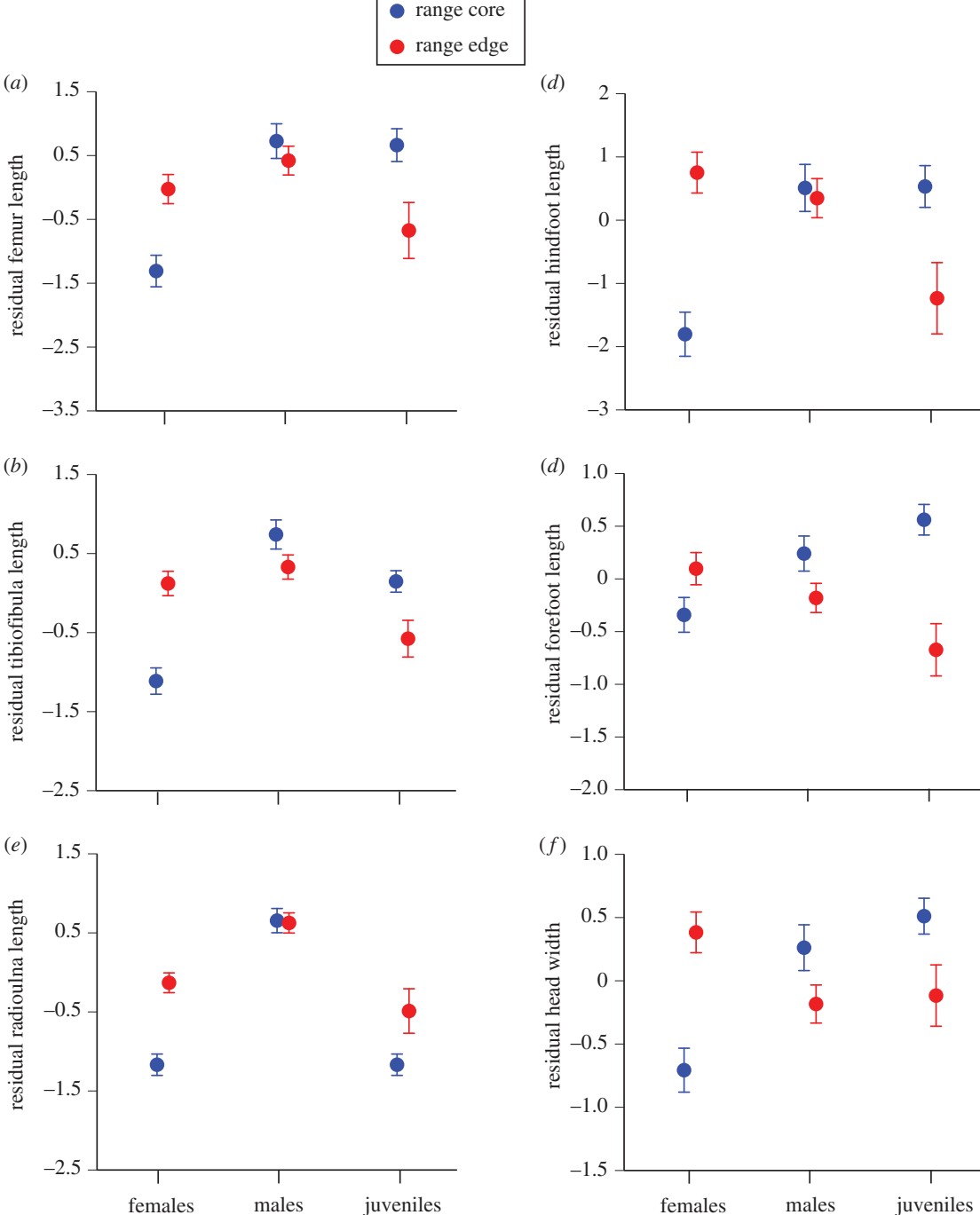

**Figure 1.** (a–f) Effects of sex/age class and geographical origin on morphological traits of cane toads, *R. marina*. To remove effects of differences in the absolute body size, the morphological variables are expressed as residual scores from the general linear regression of the relevant trait against body length. Data for F0 and F1 generations are combined. (Online version in colour.)

were intercorrelated. For example, higher velocity was associated with longer distances covered ($r = 0.76$) and greater height ($r = 0.54$), and steeper take-off angles generated higher ($r = 0.33$) hops.

### (d) Effect of location on jumping performance

In wild-caught adult toads, females from invasion-front populations exhibited higher maximum velocity during a jump ($p = 0.014$) than did range-core conspecifics (tables 1 and 2 and figure 2). The captive-raised female progeny of wild-caught range-core adults exhibited faster jumps (both average and maximum velocity) than did the progeny of invasion-front toads (tables 1 and 2). Likewise, captive-raised male

progeny of range-core toads showed flatter take-off angles and achieved lower heights than did captive-raised progeny of invasion-front toads (tables 1 and 2).

### (e) Correlations between morphology and jumping performance

Combining all data, multiple regression with backwards elimination suggests that the distance jumped by a toad was related to its body length (SVL $\beta = 0.52$, s.e. $= 0.08$, $t = 6.67$, $p < 0.0001$) and relative foot length ($\beta = 0.75$, s.e. $= 0.38$, $t = 2.00$, $p < 0.047$). The height of a jump was also related to the animal's relative hindfoot length ($\beta = 0.26$, s.e. $= 0.11$, $t = 2.38$, $p < 0.018$). Maximum velocities during jumps were higher for larger toads

Table 1. Sample sizes, mean values and associated standard errors for morphological and locomotor traits measured on cane toads (*Rhinella marina*) from the range core in QLD and the invasion front in WA. F0, wild-caught; F1, captive-raised.

| invasion category | range core | invasion front | range core | invasion front | range core | invasion front | range core | invasion front | range core | invasion front |
|---|---|---|---|---|---|---|---|---|---|---|
| sex | female (s.e.) | female (s.e.) | female (s.e.) | female (s.e.) | male (s.e.) | male (s.e.) | male (s.e.) | male (s.e.) | juvenile (s.e.) | juvenile (s.e.) |
| generation | F0 | F0 | F1 | F1 | F0 | F0 | F1 | F1 | F1 | F1 |
| *n* | 34 | 56 | 29 | 19 | 23 | 61 | 29 | 15 | 33 | 12 |
| *morphology* | | | | | | | | | | |
| mean SVL (mm) | 110.06 (1.34) | 116.14 (1.42) | 96.01 (0.94) | 96.24 (1.30) | 106.11 (1.57) | 109.92 (0.99) | 96.17 (0.89) | 96.05 (1.30) | 82.78 (1.03) | 82.99 (1.33) |
| mean mass (g) | 132.82 (7.56) | 139.32 (6.40) | 93.24 (3.86) | 87.47 (4.74) | 117.00 (5.95) | 116.07 (3.55) | 90.72 (3.20) | 94.20 (5.70) | 53.21 (2.26) | 56.33 (2.90) |
| mean femur (mm) | 45.85 (0.54) | 50.09 (0.72) | 39.33 (0.41) | 39.73 (0.62) | 46.33 (0.88) | 47.40 (0.47) | 41.33 (0.50) | 41.16 (0.73) | 35.58 (0.45) | 34.33 (0.73) |
| mean hindfoot (mm) | 68.36 (0.78) | 75.78 (0.93) | 63.18 (0.56) | 63.68 (0.71) | 68.79 (1.28) | 71.55 (0.66) | 64.88 (0.60) | 64.34 (0.84) | 56.73 (0.67) | 55.08 (1.20) |
| mean forefoot (mm) | 26.42 (0.34) | 28.70 (0.37) | 24.55 (0.22) | 23.89 (0.31) | 26.48 (0.42) | 27.06 (0.24) | 24.80 (0.28) | 23.85 (0.54) | 22.35 (0.27) | 21.16 (0.41) |
| mean head width (mm) | 39.54 (0.42) | 42.99 (0.54) | 35.14 (0.34) | 35.43 (0.40) | 39.14 (0.74) | 40.10 (0.34) | 36.18 (0.38) | 35.63 (0.61) | 31.88 (0.36) | 31.33 (0.61) |
| mean humerus (mm) | 36.59 (0.37) | 39.37 (0.50) | 30.65 (0.28) | 31.41 (0.45) | 38.38 (0.69) | 38.27 (0.32) | 33.60 (0.41) | 32.41 (0.63) | 28.67 (0.44) | 28.15 (0.66) |
| mean radioulna (mm) | 28.52 (0.31) | 31.53 (0.40) | 25.09 (0.20) | 25.11 (0.44) | 29.60 (0.46) | 30.44 (0.25) | 26.72 (0.28) | 26.47 (0.48) | 23.25 (0.34) | 22.36 (0.37) |
| *jumping performance* | | | | | | | | | | |
| distance jumped (cm) | 60.13 (3.14) | 67.21 (2.45) | 49.58 (3.24) | 44.16 (4.01) | 65.46 (3.82) | 68.93 (2.35) | 58.40 (3.24) | 55.80 (4.51) | 50.42 (3.04) | 49.72 (5.04) |
| height of jump (cm) | 10.73 (0.79) | 9.88 (0.62) | 9.38 (0.73) | 9.13 (0.91) | 11.61 (0.95) | 11.41 (0.59) | 10.98 (0.73) | 11.82 (1.02) | 9.01 (0.69) | 9.50 (1.14) |
| mean angle landing (°) | −22.18 (1.85) | −25.01 (1.39) | −26.28 (1.95) | −32.86 (2.17) | −23.66 (1.71) | −23.71 (1.29) | −27.27 (1.73) | −36.29 (2.52) | −25.23 (1.62) | −30.78 (1.94) |
| mean angle take-off (°) | 31.78 (1.70) | 30.99 (1.50) | 36.49 (2.01) | 31.74 (2.11) | 30.29 (2.25) | 33.34 (1.59) | 34.25 (2.01) | 41.06 (4.04) | 33.99 (1.81) | 30.11 (2.82) |
| mean average velocity (cm s⁻¹) | 188.73 (7.04) | 205.09 (7.33) | 189.74 (10.99) | 169.92 (11.75) | 211.94 (9.24) | 197.60 (7.67) | 214.24 (10.52) | 209.48 (18.37) | 199.47 (10.71) | 194.27 (16.81) |
| mean maximum velocity (cm s⁻¹) | 356.75 (10.80) | 408.93 (14.27) | 359.12 (20.80) | 305.76 (22.77) | 394.64 (18.53) | 390.19 (15.42) | 406.25 (20.37) | 395.68 (36.98) | 356.86 (20.12) | 350.95 (34.38) |

**Table 2.** Results of statistical tests for geographical variation in traits of cane toads that were raised in standard (common-garden) conditions in captivity (F1), and were the progeny of wild-caught adult toads (F0) collected either from the species's range core in QLD or invasion front in WA. Degrees of freedom (d.f.), F-ratios and p-values from ANOVAs with state (QLD versus WA) and population nested within state as factors are shown. The dependent variables were residual scores from general linear regressions of the relevant trait against body length (SVL), to correct for effects due to differences in the absolute body size. Italicized entries show p < 0.05.

| trait | effects of state (QLD versus WA) | | | | | | | | | effects of populations within state | | | | | | | | |
|---|---|---|---|---|---|---|---|---|---|---|---|---|---|---|---|---|---|---|
| | female d.f. | female F-ratio | female prob > F | male d.f. | male F-ratio | male prob > F | juvenile d.f. | juvenile F-ratio | juvenile prob > F | female d.f. | female F-ratio | female prob > F | male d.f. | male F-ratio | male prob > F | juvenile d.f. | juvenile F-ratio | juvenile prob > F |
| *morphology wild-caught toads (F0)* | | | | | | | | | | | | | | | | | | |
| residuals femur | 1,83 | 6.36 | *0.0136* | 1,77 | 2.62 | 0.1093 | | | | 5,83 | 2.11 | 0.0720 | 5,77 | 1.30 | 0.2711 | | | |
| residuals hindfoot | 1,83 | 37.18 | *<0.0001* | 1,77 | 0.28 | 0.5988 | | | | 5,83 | 1.91 | 0.1014 | 5,77 | 1.07 | 0.3836 | | | |
| residuals forefoot | 1,83 | 6.13 | *0.0153* | 1,77 | 1.03 | 0.3123 | | | | 5,83 | 1.45 | 0.2160 | 5,77 | 1.88 | 0.1069 | | | |
| residuals head width | 1,83 | 14.28 | *0.0003* | 1,77 | 2.32 | 0.1316 | | | | 5,83 | 1.25 | 0.2914 | 5,77 | 2.07 | 0.0786 | | | |
| residuals tibiofibula | 1,83 | 13.09 | *0.0005* | 1,77 | 5.35 | *0.0234* | | | | 5,83 | 3.36 | *0.0082* | 5,77 | 2.15 | 0.0682 | | | |
| residuals radioulna | 1,83 | 20.84 | *<0.0001* | 1,77 | 2.29 | 0.1347 | | | | 5,83 | 7.07 | *<0.0001* | 5,77 | 1.76 | 0.1319 | | | |
| *morphology captive-raised progeny (F1)* | | | | | | | | | | | | | | | | | | |
| residuals femur | 1,41 | 2.63 | 0.1126 | 1,38 | 1.79 | 0.1888 | 1,38 | 8.17 | *0.0069* | 5,41 | 1.11 | 0.3677 | 4,38 | 1.42 | 0.2450 | 5,38 | 1.20 | 0.3297 |
| residuals hindfoot | 1,41 | 0.51 | 0.4783 | 1,38 | 1.07 | 0.3069 | 1,38 | 6.45 | *0.0153* | 5,41 | 0.34 | 0.8884 | 4,38 | 0.37 | 0.8297 | 5,38 | 0.84 | 0.5324 |
| residuals forefoot | 1,41 | 3.32 | 0.0756 | 1,38 | 5.83 | *0.0207* | 1,38 | 17.70 | *0.0002* | 5,41 | 1.02 | 0.4188 | 4,38 | 3.20 | *0.0232* | 5,38 | 1.92 | 0.1143 |
| residuals head width | 1,41 | 7.72 | *0.0082* | 1,38 | 0.69 | 0.4117 | 1,38 | 4.47 | *0.0411* | 5,41 | 4.64 | *0.0019* | 4,38 | 4.26 | *0.0061* | 5,38 | 1.64 | 0.1735 |
| residuals tibiofibula | 1,41 | 7.33 | *0.0098* | 1,38 | 1.43 | 0.2387 | 1,38 | 8.12 | *0.0070* | 5,41 | 2.38 | 0.0549 | 4,38 | 0.79 | 0.5395 | 5,38 | 3.85 | *0.0064* |
| residuals radioulna | 1,41 | 0.39 | 0.5353 | 1,38 | 1.52 | 0.2255 | 1,38 | 9.37 | *0.0040* | 5,41 | 2.89 | *0.0253* | 4,38 | 1.76 | 0.1574 | 5,38 | 0.88 | 0.5027 |
| *locomotor performance wild-caught toads (F0)* | | | | | | | | | | | | | | | | | | |
| distance jump | 1,83 | 2.31 | 0.1325 | 1,77 | 0.66 | 0.4202 | | | | 5,83 | 2.48 | *0.0382* | 5,77 | 5.67 | *0.0002* | | | |
| height jump | 1,83 | 2.38 | 0.1269 | 1,77 | 0.52 | 0.4748 | | | | 5,83 | 1.54 | 0.1865 | 5,77 | 2.49 | *0.0380* | | | |
| average velocity | 1,83 | 2.47 | 0.1197 | 1,77 | 0.90 | 0.3460 | | | | 5,83 | 2.60 | *0.0310* | 5,77 | 2.67 | *0.0285* | | | |
| maximum velocity | 1,83 | 6.25 | *0.0144* | 1,77 | 0.37 | 0.5425 | | | | 5,83 | 1.37 | 0.2440 | 5,77 | 3.09 | *0.0139* | | | |
| angle take-off | 1,83 | 0.46 | 0.4999 | 1,77 | 0.22 | 0.6388 | | | | 5,83 | 3.51 | *0.0063* | 5,77 | 0.54 | 0.7457 | | | |
| angle landing | 1,83 | 0.69 | 0.4091 | 1,77 | 0.00 | 0.9713 | | | | 5,83 | 0.93 | 0.4673 | 5,77 | 0.75 | 0.5909 | | | |
| *locomotor performance captive-raised progeny (F1)* | | | | | | | | | | | | | | | | | | |
| distance jumped | 1,41 | 1.93 | 0.1725 | 1,38 | 0.22 | 0.6447 | 1,38 | 0.01 | 0.9382 | 5,41 | 0.29 | 0.9138 | 4,38 | 1.17 | 0.3397 | 5,38 | 0.89 | 0.4991 |
| height of jump | 1,41 | 0.04 | 0.8366 | 1,38 | 4.53 | *0.0398* | 1,38 | 0.07 | 0.7924 | 5,41 | 0.50 | 0.7760 | 4,38 | 2.94 | *0.0327* | 5,38 | 1.31 | 0.2796 |
| average velocity | 1,41 | 4.30 | *0.0445* | 1,38 | 0.55 | 0.4631 | 1,38 | 0.00 | 0.9513 | 5,41 | 0.90 | 0.4896 | 4,38 | 1.18 | 0.3337 | 5,38 | 0.44 | 0.8191 |
| maximum velocity | 1,41 | 5.41 | *0.0250* | 1,38 | 2.98 | 0.0925 | 1,38 | 0.01 | 0.9306 | 5,41 | 0.76 | 0.5815 | 4,38 | 3.04 | *0.0292* | 5,38 | 0.62 | 0.6839 |
| angle take-off | 1,41 | 3.06 | 0.0877 | 1,38 | 5.96 | *0.0196* | 1,38 | 0.98 | 0.3275 | 5,41 | 0.63 | 0.6814 | 4,38 | 1.50 | 0.2235 | 5,38 | 0.86 | 0.5189 |
| angle landing | 1,41 | 3.16 | 0.0828 | 1,38 | 3.79 | 0.0591 | 1,38 | 3.01 | 0.0907 | 5,41 | 0.60 | 0.6987 | 4,38 | 0.21 | 0.9306 | 5,38 | 0.22 | 0.9533 |

Proc. R. Soc. B 287: 20201964

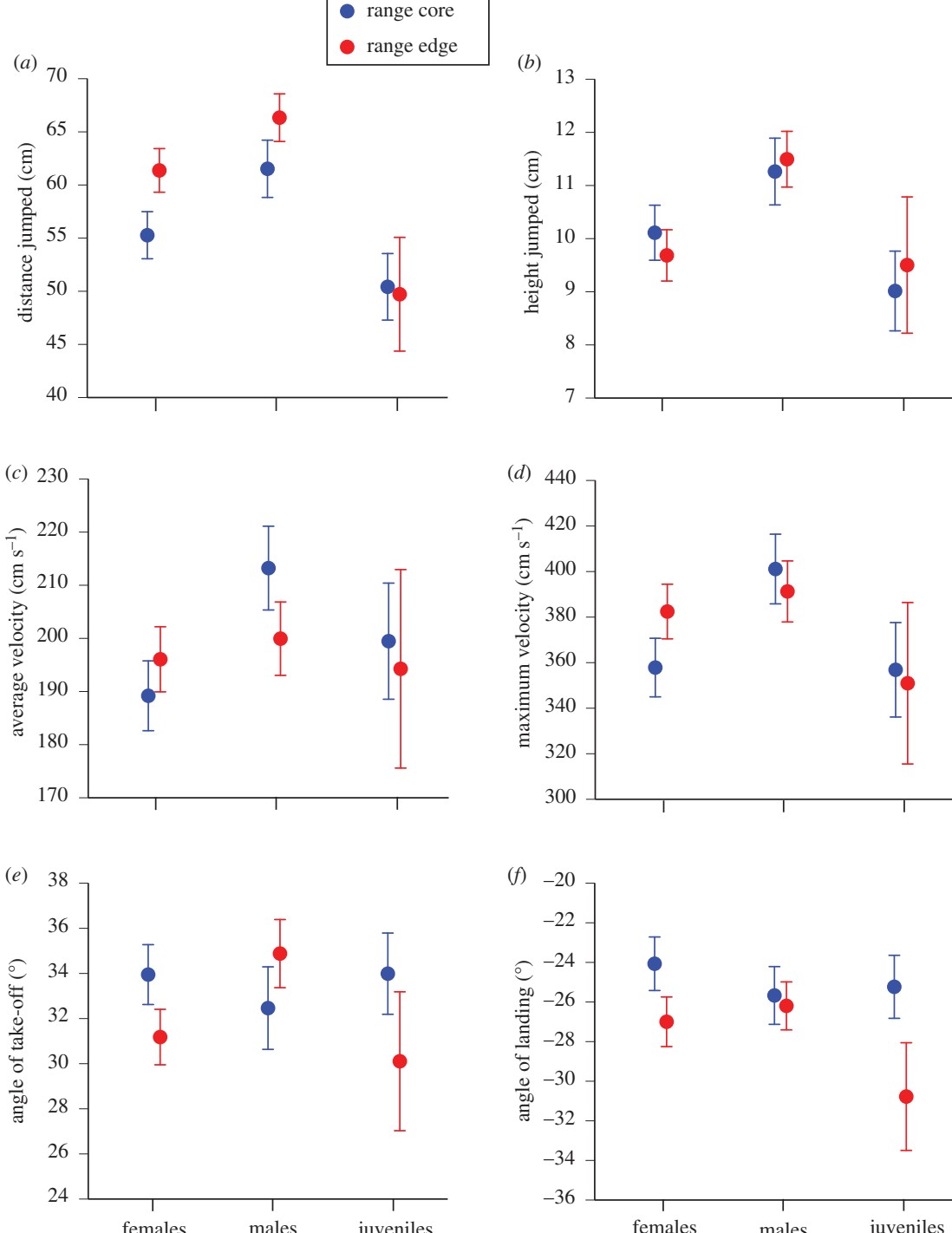

**Figure 2.** (a–f) Effects of sex/age class and geographical origin on locomotor traits of cane toads, R. marina, as tested in jumping trials and raceway trials. (Online version in colour.)

(SVL $\beta = 1.25$, s.e. $= 0.46$, $t = 2.72$, $p = 0.007$) and for those with relatively longer hindfeet ($\beta = 5.89$, s.e. $= 2.22$, $t = 2.66$, $p = 0.008$). Larger toads also had shallower angles of landing ($\beta = 0.10$, s.e. $= 0.04$, $t = 2.15$, $p = 0.03$).

If we look only at adult females (the group with most geographical variation in both morphology and performance; table 2), pairwise correlations on data for wild-caught toads showed that steeper angles of landing were associated with relatively narrow heads ($r = -0.24$, $n = 90$, $p < 0.022$) and shorter femurs ($r = -0.25$, $n = 90$, $p < 0.016$). In captive-reared female offspring, a longer radioulna was associated with longer hops and greater mean speeds in raceway trials (respectively, $r = 0.51$, $n = 19$, $p < 0.025$; $r = 0.53$, $n = 19$, $p < 0.019$).

## (f) Comparisons between morphology and raceway performance

Our two measures of raceway performance (speed and average distance per hop) were themselves highly correlated ($n = 89$, $r = 0.67$, $p < 0.0001$) and were linked to several of the residual scores for morphology that we calculated. Using multiple regression with backwards elimination (as above), we found that the mean distance covered per hop in raceway trials was positively related to a toad's body length ($\beta = 0.002$, s.e. $= 0.0003$, $t = 5.62$, $p < 0.0001$), relative forefoot length ($\beta = 0.01$, s.e. $= 0.005$, $t = 2.13$, $p = 0.036$) and tibiofibula length ($\beta = 0.01$, s.e. $= 0.004$, $t = 3.15$, $p = 0.002$), and negatively

related to its relative head width ($\beta = -0.01$, s.e. $= 0.0004$, $t = -2.26$, $p = 0.027$). Speed on the raceway was also positively related to a toad's body size ($\beta = 0.001$, s.e. $= 0.0008$, $t = 2.32$, $p = 0.024$) and relative forefoot length ($\beta = 0.03$, s.e. $= 0.01$, $t = 2.33$, $p = 0.0022$).

## (g) Comparisons between jumping performance and raceway performance

For the 89 individuals for which we have data both on jumping performance and raceway performance, we can compare those two datasets. Toads that travelled further per hop in the raceway also travelled further per jump ($r = 0.32$, $p < 0.0022$) and jumped higher ($r = 0.22$, $p < 0.04$) in our locomotor video trials.

## 4. Discussion

Variation in morphology and locomotor performance was linked in the cane toads that we studied, and those suites of traits differed between toads from range-core versus invasion-front populations. Relative to body length, toads from the invasion front had longer limb bones (femur, tibiofibula, radioulna), larger hindfeet and smaller forefeet. Those divergences were greater in adult females than in other toads (table 2). The invasion-front animals also tended to jump differently than their range-core counterparts, although differences for many aspects of jumping performance were minor (table 2 and figure 2). Comparisons between data from raceway trials and jumping trials show that measurements of variation in distances per hop can predict 10% of the variation in locomotor performance over longer distances (15 m).

Previous research has documented divergence between range-core and invasion-front toads in other aspects of locomotor performance. Importantly, cane toads from the native range in South America resemble those from the range core in Australia, indicating that the phenotypic traits of invasion-front toads (at least for morphology) represent a derived not ancestral condition [24]. In the field, invasion-front toads dispersed further per day than did range-core conspecifics [21,22] and the rate of dispersal declined rapidly in the years following the toads' initial arrival at a site [39]. By forcing toads to travel along raceways, Llewelyn et al. [40] found three-fold higher endurance (distances moved) in wild-caught invasion-front toads than in range-core conspecifics. However, Tracy et al. [41] reported no significant difference in a similar study, based on animals that had been held in captivity for a longer period prior to testing. Toads from the invasion front were more adept at climbing than were toads from the range core [24]. Unlike range-core conspecifics, invasion-front toads tended to move in consistent directions rather than meandering [28]. In sum, dispersal-relevant traits have shifted profoundly during the toads' Australian invasion.

Importantly, the geographical divergences in our overall dataset were also evident (and in some cases stronger) when we looked only at offspring raised under standard (common-garden) conditions in captivity (table 2). Their enclosures gave no opportunity for sustained locomotion, so that the longer legs and greater athleticism of invasion-front animals cannot be attributed to phenotypically plastic responses to higher rates of dispersal at the invasion front (see [42]). In keeping with our results, Hudson et al. [24]

and Stuart et al. [43] reported that relative leg lengths differed among progeny of toads from different populations. In the Stuart et al. [43] study, exercising the young toads did not affect their relative leg lengths (but see [44] for a counter-example with lizards). Captive-raised progeny of range-core and invasion-front toads also exhibited significant differences in stamina, and in the effects of rearing conditions on stamina [43]. Heritable differences in stamina of adult toads may involve shifts in thermal effects on this trait, with invasion-front individuals outperforming range-core individuals only when the animals are tested at high temperatures [26]. At least some dispersal-relevant behaviours are heritable [5,28].

Previous studies on cane toads also have documented links between morphology and locomotion. Longer-legged toads tended to travel further per week in the field, and to move more quickly in standardized trials of escape ability [31]. Longer legs also assisted toads to climb [23] and in laboratory raceway trials, longer-legged toads were more willing to run [32]. The analyses in the present paper add to this body of evidence, in that we found strong correlations between a toad's morphology and its performance both in jumping trials and in raceway trials. Importantly, those form-function correlations support the idea that morphological modifications that have arisen during the toad's invasion of Australia have been accompanied by shifts in locomotor performance, as inferred but not documented by earlier studies [31]. Morphological traits such as relative length of forelimbs or relative limb length ratio (forelimb length/hindlimb length) also might be correlated with habitat usage and locomotor kinematics in cane toads [45,46], as these morphological traits play a crucial role during the landing phase [47]. Moreover, forelimb length and relative limb length ratio might influence the ability of cane toads to perform bouts of repeated hopping [48].

However, we cannot dismiss the possibility that the correlations between morphology and locomotor ability might be due to undetected effects of other variables. For example, invasion-front toads tend to be bolder and more active than range-core conspecifics [29,30]. That behavioural difference, rather than morphological adaptations, might modify their performance in jumping trials. However, the plausibility of this interpretation is weakened by the strong concordance between our results and the predictions from biomechanical models: for example, longer limbs are expected to provide greater propulsive power, and hence to generate longer jumps [8,33,49–51]. This is broadly the pattern that we have seen. The shifts in foot morphology, and their correlations with jumping performance, are more ambiguous. Increased foot length may be an adaptive consequence of the transition to more powerful jumps, but also may reflect a greater reliance on bounding (repeated rather than single hops) in range-front individuals [24]. In toads, bounding locomotion, with no resting phase between successive hops, requires modified placement of the feet during take-off and landing, and allows for efficient transfer of elastic energy between hops through the forearms [47,48].

Although all age/sex classes showed morphological and locomotor divergence among sites, those divergences tended to be stronger for females than for males or juveniles (figures 1 and 2, and tables 1 and 2). Why should females be more strongly affected by the invasion process? Mathematical models suggest that if the sexes differ in intrinsic capacity for dispersal during an accelerating invasion, selection for rapid

dispersal will be stronger on the slower sex than the faster sex (because the latter otherwise will outrun all of their potential sexual partners [52]). In cane toads from the range core in Australia, males have relatively longer legs (figure 1a–c) and are faster than females of the same body size (figure 2d), a pattern also seen in other toad species [53]. Perhaps as a result, overall leg length (across both sexes) has converged to more female-like dimensions in both sexes over the course of the cane toad invasion [24]. Also, male limb lengths are under strong sexual selection in this species: for example, shorter forelimbs enable an amplexing male to cling more strongly to a female [54,55]. Those conflicting advantages for dispersal in some situations but not others, and for success in sexual struggles, may have weakened net selection on limb lengths and mobility in male toads.

In summary, our study provides empirical evidence for an oft-assumed but rarely tested assumption: the idea that 'dispersal-enhancing' traits at an invasion front do actually affect locomotor performance. Often, investigators have interpreted distinctive phenotypic traits in invasion-front individuals as adaptations to accelerate movement, but without performance data. For example, voles that reached outlying Swedish islands had longer feet, plausibly enabling them to swim more effectively [18]. Bush crickets at an invasion-front in Britain had larger wings, putatively enhancing flight ability [17]. Pine trees at the expanding northern range edge had lighter seeds, presumably enabling them to travel further on the wind [16]. We cite these examples not as criticisms, but to point out that logistical constraints often prevent investigators (ourselves included) from fully exploring the assumptions inherent in their interpretations. Encouragingly, our data on cane toads show the patterns that we expect, and importantly, show those patterns not only in wild-caught animals, but in their captive-raised (common-garden) progeny. As cane toads have spread across Australia, they have evolved not only in body shape but also in the ways that they move.

Ethics. All procedures were approved by the University of Sydney Animal Care and Ethics Committee (Protocol no. 6705).

Data accessibility. Data are available from the Dryad Data Repository: https://dx.doi.org/10.5061/dryad.pk0p2ngkw [56]. Code associated with our kinematics pipeline has been deposited in the GitHub repository: https://github.com/marta-vidalgarcia/jumping-toads.

Authors' contributions. R.S., C.M.H. and M.V.-G. conceived the study. C.M.H. and M.V.-G. gathered the data, and all authors contributed to data analysis and manuscript preparation.

Competing interests. The authors declare no competing interests.

Funding. This work was funded by the Australian Research Council (FL120100074).

Acknowledgements. We thank Georgia Ward-Fear, Greg Brown and Jodie Gruber for assistance with toad collection; Scott Keogh for advice and encouragement; Melanie Elphick for formatting the manuscript and Simon Baeckens for comments on the manuscript.

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
