## [Reviewer comments · Proceedings of the Royal Society B: Biological Sciences]

Review History

RSPB-2020-1964.R0 (Original submission)

Review form: Reviewer 1

Recommendation

Accept with minor revision (please list in comments)

Scientific importance: Is the manuscript an original and important contribution to its field?

Excellent

General interest: Is the paper of sufficient general interest?

Excellent

Quality of the paper: Is the overall quality of the paper suitable?

Excellent

Is the length of the paper justified?

Yes

Should the paper be seen by a specialist statistical reviewer?

No

Do you have any concerns about statistical analyses in this paper? If so, please specify them explicitly in your report.

No

It is a condition of publication that authors make their supporting data, code and materials available - either as supplementary material or hosted in an external repository. Please rate, if applicable, the supporting data on the following criteria.

Is it accessible?

Yes

Is it clear?

Yes

Is it adequate?

Yes

Do you have any ethical concerns with this paper?

No

Comments to the Author

This is a significant study of likely wide interest to the readership of this journal.

For me, the key finding is that females at the range edge in F0 and F1 generations jump relatively far compared with females at the core. Whilst all measures of size were greater in range edge than core females in F0, only head width and tibiofibular length persisted as larger in F1. This is a great set of data, but I think it would have been very useful in results section (f) to have considered females separately as well, rather than just the various grouping that were used. Essentially the females are the interesting group and it would be good to see whether variation in measures of body size predict variation in performance of females. The discussion could then include more specific references to the results from females. The results, including the figures, tend analyse various groups together thereby masking the interesting findings in the females. The later discussion rightly picks up the contrast in findings between male and female groups.

Minor comments

Line 99 I suggest changing '(see references above)' to something along the lines of '(see introduction and references cited therein)'.

Line 106 It would be useful to know how long these individuals were maintained in captivity and under what conditions. Similarly it would be useful to know how long the F1 individuals (line 123) were raised for and under what conditions.

Line 186 I wonder whether you considered using principal component analysis to remove any issues in collinearity between independent variables, such as the various morphometric variables

Line 224: Table 2 suggests that male wild caught differ between range edge and core in only tibiofibular, not radioulna.

Line 241 The statement here does not match Table 2. In Table 2 there are no significant differences in locomotor performance between edge and core in males.

Line 245 Are you sure that is correct? In theory, steeper take-off angles would be more likely to achieve greater heights.

Lines 282-284 The sentence starting ‘The invasion-front animals also tended to jump differently...’ could be more specific. In what way do they differ?

Lines 285-286 I’m not sure I would agree with this statement. If $r = 0.32$, then $r^2 = 0.10$, so 90% of the variation in locomotor performance over longer distances is unexplained by this relationship. You could alter ‘measurements of distances per hop can predict locomotor performance over longer distances (15 metres).’ To ‘measurements of variation in distance per hop can predict 10% of the variation in locomotor performance over longer distances (15 metres).’ If you explain in the methods what you are trying to do with r^2 calculations.

Lines 293-295 This seems a key finding to me. Endurance seems a likely key factor in rapid dispersal, more so than being able to jump far in one jump or quickly over 15m. Endurance is also likely highly trainable, although a genetic component is also likely.

Line 383 Author contributions statement: CMH referred to as CMM.

Table 1: I wonder whether it would be helpful to also express some of the jumping performance variables in terms of SVL to account for variation in body size between groups.

Review form: Reviewer 2 (Alan Savitzky)

Recommendation

Accept with minor revision (please list in comments)

Scientific importance: Is the manuscript an original and important contribution to its field?

Excellent

General interest: Is the paper of sufficient general interest?

Excellent

Quality of the paper: Is the overall quality of the paper suitable?

Excellent

Is the length of the paper justified?

Yes

Should the paper be seen by a specialist statistical reviewer?

No

Do you have any concerns about statistical analyses in this paper? If so, please specify them explicitly in your report.

No

It is a condition of publication that authors make their supporting data, code and materials available - either as supplementary material or hosted in an external repository. Please rate, if applicable, the supporting data on the following criteria.

Is it accessible?

Yes

Is it clear?

Yes

Is it adequate?

Yes

Do you have any ethical concerns with this paper?

No

Comments to the Author

This manuscript adds an important dimension to the well-documented account of evolution in the invasive Australian population of *Rhinella marina*. As the authors point out, the results of this study are not surprising, but their very substantial value derives from the fact that locomotor performance was carefully and explicitly tested in a manner that permits confirmation of the expectations drawn previously from morphology alone. The manuscript is also of value for the methods employed to record and analyze toad locomotion in three dimensions, which are carefully described here.

I would note several minor points:

Line 122: The level of precision achievable using Vernier calipers should be noted (e.g., "... to the nearest 0.XX mm").

Line 153: Delete comma, to read "information through"

Lines 224, 227, 264: Change "tibiofibular" (an adjective, as in "tibiofibular length") to "tibiofibula" (a noun). The term is spelled correctly on Line 281 and in Table 2.

Decision letter (RSPB-2020-1964.R0)

23-Sep-2020

Dear Professor Shine:

Your manuscript has now been peer reviewed and the reviews have been assessed by an Associate Editor. The reviewers' comments (not including confidential comments to the Editor) and the comments from the Associate Editor are included at the end of this email for your reference. As you will see, the reviewers and the Editors have raised some concerns with your manuscript and we would like to invite you to revise your manuscript to address them.

Research ethics:

Use of animals and field studies:

It is a condition of publication that you make available the data and research materials supporting the results in the article. Please see our Data Sharing Policies (<https://royalsociety.org/journals/authors/author-guidelines/#data>). Datasets should be deposited in an appropriate publicly available repository and details of the associated accession number, link or DOI to the datasets must be included in the Data Accessibility section of the article (<https://royalsociety.org/journals/ethics-policies/data-sharing-mining/>). Reference(s) to datasets should also be included in the reference list of the article with DOIs (where available).

If you wish to submit your data to Dryad (<http://datadryad.org/>) and have not already done so you can submit your data via this link [http://datadryad.org/submit?journalID=RSPB&manu=\(Document not available\)](http://datadryad.org/submit?journalID=RSPB&manu=(Document%20not%20available)), which will take you to your unique entry in the Dryad repository.

Please submit a copy of your revised paper within three weeks. If we do not hear from you within this time your manuscript will be rejected. If you are unable to meet this deadline please let us know as soon as possible, as we may be able to grant a short extension.

Best wishes,
Dr John Hutchinson, Editor
mailto: proceedingsb@royalsociety.org

Associate Editor
Board Member: 1
Comments to Author:

Dear Dr. Shine,

Thank you for submitting your manuscript entitled "The accelerating anuran: evolution of locomotor performance in cane toads (*Rhinella marina*, Bufonidae) at an invasion front" to the Proceedings of the Royal Society. I have received two peer reviews, and both are highly supportive of your manuscript but also have a few suggestions, which I hope you will find useful when revising your manuscript.

I appreciate that your manuscript examines the locomotor performance of cane toads, comparing toads at the invasion front with those in the range core and providing insights into how locomotor performance might drive the invasive spread of cane toads in Australia.

Proceedings B aims to publish studies that significantly increase or alter our current understandings in a way that is relevant to a broad readership beyond the disciplinary area of the manuscript. Both reviewers find your study of excellent scientific importance and broad interest and many of their comments aim mainly at improving the clarity of the manuscript's arguments.

Reviewer 1 furthermore suggests that the study should include a more focused analysis and discussion of the data on female toads, a suggestion which I encourage you to consider if the data allow such a statistical treatment.

Reviewer(s)' Comments to Author:

Referee: 1

Comments to the Author(s)

This is a significant study of likely wide interest to the readership of this journal.

For me, the key finding is that females at the range edge in F0 and F1 generations jump relatively far compared with females at the core. Whilst all measures of size were greater in range edge than core females in F0, only head width and tibiofibular length persisted as larger in F1. This is a great set of data, but I think it would have been very useful in results section (f) to have considered females separately as well, rather than just the various grouping that were used.

Essentially the females are the interesting group and it would be good to see whether variation in measures of body size predict variation in performance of females. The discussion could then include more specific references to the results from females. The results, including the figures, tend analyse various groups together thereby masking the interesting findings in the females. The later discussion rightly picks up the contrast in findings between male and female groups.

Minor comments

Line 99 I suggest changing '(see references above)' to something along the lines of '(see introduction and references cited therein)'.

Line 106 It would be useful to know how long these individuals were maintained in captivity and under what conditions. Similarly it would be useful to know how long the F1 individuals (line 123) were raised for and under what conditions.

Line 186 I wonder whether you considered using principal component analysis to remove any issues in collinearity between independent variables, such as the various morphometric variables

Line 224: Table 2 suggests that male wild caught differ between range edge and core in only tibiofibular, not radioulna.

Line 241 The statement here does not match Table 2. In Table 2 there are no significant differences in locomotor performance between edge and core in males.

Line 245 Are you sure that is correct? In theory, steeper take-off angles would be more likely to achieve greater heights.

Lines 282-284 The sentence starting 'The invasion-front animals also tended to jump differently...' could be more specific. In what way do they differ?

Lines 285-286 I'm not sure I would agree with this statement. If $r = 0.32$, then $r^2 = 0.10$, so 90% of the variation in locomotor performance over longer distances is unexplained by this relationship. You could alter 'measurements of distances per hop can predict locomotor performance over longer distances (15 metres).' To 'measurements of variation in distance per hop can predict 10% of the variation in locomotor performance over longer distances (15 metres).' If you explain in the methods what you are trying to do with r^2 calculations.

Lines 293-295 This seems a key finding to me. Endurance seems a likely key factor in rapid dispersal, more so than being able to jump far in one jump or quickly over 15m. Endurance is also likely highly trainable, although a genetic component is also likely.

Line 383 Author contributions statement: CMH referred to as CMM.

Table 1: I wonder whether it would be helpful to also express some of the jumping performance variables in terms of SVL to account for variation in body size between groups.

Referee: 2

Comments to the Author(s)

This manuscript adds an important dimension to the well-documented account of evolution in the invasive Australian population of *Rhinella marina*. As the authors point out, the results of this study are not surprising, but their very substantial value derives from the fact that locomotor performance was carefully and explicitly tested in a manner that permits confirmation of the expectations drawn previously from morphology alone. The manuscript is also of value for the methods employed to record and analyze toad locomotion in three dimensions, which are carefully described here.

I would note several minor points:

Line 122: The level of precision achievable using Vernier calipers should be noted (e.g., "... to the nearest 0.XX mm").

Line 153: Delete comma, to read "information through"

Lines 224, 227, 264: Change "tibiofibular" (an adjective, as in "tibiofibular length") to "tibiofibula" (a noun). The term is spelled correctly on Line 281 and in Table 2.

Author's Response to Decision Letter for (RSPB-2020-1964.R0)

See Appendix A.

RSPB-2020-1964.R1 (Revision)

Review form: Reviewer 1 (Rob James)

Recommendation

Accept as is

Scientific importance: Is the manuscript an original and important contribution to its field?

Good

General interest: Is the paper of sufficient general interest?

Excellent

Quality of the paper: Is the overall quality of the paper suitable?

Good

Is the length of the paper justified?

Yes

Should the paper be seen by a specialist statistical reviewer?

No

Do you have any concerns about statistical analyses in this paper? If so, please specify them explicitly in your report.

No

It is a condition of publication that authors make their supporting data, code and materials available - either as supplementary material or hosted in an external repository. Please rate, if applicable, the supporting data on the following criteria.

Is it accessible?

Yes

Is it clear?

Yes

Is it adequate?

Yes

Do you have any ethical concerns with this paper?

No

Comments to the Author

I am happy with the revisions made to this paper

Decision letter (RSPB-2020-1964.R1)

19-Oct-2020

Dear Professor Shine

I am pleased to inform you that your manuscript entitled "The accelerating anuran: evolution of locomotor performance in cane toads (*Rhinella marina*, Bufonidae) at an invasion front" has been accepted for publication in Proceedings B. Congratulations!!

Open Access

Your article has been estimated as being 9 pages long. Our Production Office will be able to confirm the exact length at proof stage.

Paper charges

Sincerely,

Dr John Hutchinson

Appendix A

Author response letter

Hudson et al., locomotion in toads

****Below, we have pasted in all of the editorial and reviewer comments, and explained our responses below each specific point.**

Editorial email of 23-Sep-2020

Dear Professor Shine:

Your manuscript has now been peer reviewed and the reviews have been assessed by an Associate Editor. The reviewers' comments (not including confidential comments to the Editor) and the comments from the Associate Editor are included at the end of this email for your reference. As you will see, the reviewers and the Editors have raised some concerns with your manuscript and we would like to invite you to revise your manuscript to address them.

Research ethics:

Use of animals and field studies:

****We have moved the information about ethics approval to the Methods section.**

It is a condition of publication that you make available the data and research materials supporting the results in the article. Please see our Data Sharing Policies (<https://royalsociety.org/journals/authors/author-guidelines/#data>). Datasets should be deposited in an appropriate publicly available repository and details of the associated accession number, link or DOI to the datasets must be included in the Data Accessibility section of the article (<https://royalsociety.org/journals/ethics-policies/data-sharing-mining/>). Reference(s) to datasets should also be included in the reference list of the article with DOIs (where available).

If you wish to submit your data to Dryad (<http://datadryad.org/>) and have not already done so you can submit your data via this link <http://datadryad.org/submit?journalID=RSPB&manu=>(Document not available), which will take you to your unique entry in the Dryad repository.

****We have now uploaded the datafiles to Dryad, and provide the reference in the manuscript.**

Online supplementary material will also carry the title and description provided during submission, so please ensure these are accurate and informative. Note that the Royal Society will not edit or typeset supplementary material and it will be hosted as provided. Please ensure that the supplementary material includes the paper details (authors, title, journal

name, article DOI). Your article DOI will be 10.1098/rspb.[paper ID in form xxxx.xxxx e.g. 10.1098/rspb.2016.0049].

Please submit a copy of your revised paper within three weeks. If we do not hear from you within this time your manuscript will be rejected. If you are unable to meet this deadline please let us know as soon as possible, as we may be able to grant a short extension.

Best wishes,

Dr John Hutchinson, Editor
mailto: proceedingsb@royalsociety.org

Associate Editor

Board Member: 1

Comments to Author:

Dear Dr. Shine,

*Thank you for submitting your manuscript entitled “The accelerating anuran: evolution of locomotor performance in cane toads (*Rhinella marina*, *Bufo*idae) at an invasion front” to the Proceedings of the Royal Society. I have received two peer reviews, and both are highly supportive of your manuscript but also have a few suggestions, which I hope you will find useful when revising your manuscript.*

I appreciate that your manuscript examines the locomotor performance of cane toads, comparing toads at the invasion front with those in the range core and providing insights into how locomotor performance might drive the invasive spread of cane toads in Australia.

Proceedings B aims to publish studies that significantly increase or alter our current understandings in a way that is relevant to a broad readership beyond the disciplinary area of the manuscript. Both reviewers find your study of excellent scientific importance and broad interest and many of their comments aim mainly at improving the clarity of the manuscript’s arguments. Reviewer 1 furthermore suggests that the study should include a more focused analysis and discussion of the data on female toads, a suggestion which I encourage you to consider if the data allow such a statistical treatment.

****Thank you for considering our manuscript for publication in Proceedings of the Royal Society, and for providing such a quick turnaround time. We have considered the reviewers’ comments, and revised the manuscript accordingly (see below). They were very helpful. We have also expanded upon the analysis and discussion of female toad data, as suggested by Reviewer 1.**

Reviewer(s)’ Comments to Author:

Referee: 1

Comments to the Author(s)

This is a significant study of likely wide interest to the readership of this journal. For me, the key finding is that females at the range edge in F0 and F1 generations jump relatively far compared with females at the core. Whilst all measures of size were greater in range edge than core females in F0, only head width and tibiofibular length persisted as larger in F1. This is a great set of data, but I think it would have been very useful in results section (f) to have considered females separately as well, rather than just the various grouping that were used. Essentially the females are the interesting group and it would be good to see whether variation in measures of body size predict variation in performance of females. The discussion could then include more specific references to the results from females. The results, including the figures, tend analyse various groups together thereby masking the interesting findings in the females. The later discussion rightly picks up the contrast in findings between male and female groups.

****We have now looked at associations between morphology and locomotor traits in females only, and report the significant patterns (for both wild-caught and common-garden specimens) in the text.**

Minor comments

Line 99 I suggest changing '(see references above)' to something along the lines of '(see introduction and references cited therein)'.

****Changed as suggested.**

Line 106: It would be useful to know how long these individuals were maintained in captivity and under what conditions. Similarly it would be useful to know how long the F1 individuals (line 123) were raised for and under what conditions.

****The text explains that the wild-caught animals were collected in October to December in 2013, and tested in October 2014/March 2016. We have added additional information to say that the common-garden offspring were 19 to 24 months of age when tested. Cane toads typically reach maturity between 9-12 months of age, so while there was a difference in mean ages between some families, all F1 individuals had reached adulthood by the time of testing.**

Line 186 I wonder whether you considered using principal component analysis to remove any issues in collinearity between independent variables, such as the various morphometric variables

****Yes, we initially ran PCAs but they did not prove useful – we were left with almost as many axes as we had original variables.**

Line 224: Table 2 suggests that male wild caught differ between range edge and core in only tibiofibular, not radioulna.

****Thanks for noticing this error, now corrected.**

Line 241 The statement here does not match Table 2. In Table 2 there are no significant

differences in locomotor performance between edge and core in males.

****Thanks for noticing this error, now corrected.**

Line 245 Are you sure that is correct? In theory, steeper take-off angles would be more likely to achieve greater heights.

***Thanks for noticing this typo. We went back and redid the tests. As the reviewer suggests, the range-core males showed flatter (not steeper) take-off angles associated with smaller maximum heights.**

Lines 282-284 The sentence starting 'The invasion-front animals also tended to jump differently...' could be more specific. In what way do they differ?

****We cite the relevant Table and Figure here, and don't see any way to succinctly summarise the nature of differences in locomotor traits between range-core and range-edge animals – because those differences shift among traits and age/sex classes. Thus, we have not made a change here.**

Lines 285-286 I'm not sure I would agree with this statement. If $r = 0.32$, then $r^2 = 0.10$, so 90% of the variation in locomotor performance over longer distances is unexplained by this relationship. You could alter 'measurements of distances per hop can predict locomotor performance over longer distances (15 metres).' To 'measurements of variation in distance per hop can predict 10% of the variation in locomotor performance over longer distances (15 metres).' If you explain in the methods what you are trying to do with r^2 calculations.

****Yes, we agree – now changed as suggested.**

Lines 293-295 This seems a key finding to me. Endurance seems a likely key factor in rapid dispersal, more so than being able to jump far in one jump or quickly over 15m. Endurance is also likely highly trainable, although a genetic component is also likely.

****We agree with this statement; endurance likely is a key trait.**

Line 383 Author contributions statement: CMH referred to as CMM.

****Oops! Thanks for noticing, now corrected.**

Table 1: I wonder whether it would be helpful to also express some of the jumping performance variables in terms of SVL to account for variation in body size between groups.

****The critical comparisons here are within age-sex groups but between populations, not between age-sex groups. We are not really interested in how males and females differ, etc., but in how traits have changed within a group over the course of the invasion. The way we express this achieves that aim.**

Referee: 2

Comments to the Author(s)

*This manuscript adds an important dimension to the well-documented account of evolution in the invasive Australian population of *Rhinella marina*. As the authors point out, the results of this study are not surprising, but their very substantial value derives from the fact that locomotor performance was carefully and explicitly tested in a manner that permits*

confirmation of the expectations drawn previously from morphology alone. The manuscript is also of value for the methods employed to record and analyze toad locomotion in three dimensions, which are carefully described here.

I would note several minor points:

Line 122: The level of precision achievable using Vernier calipers should be noted (e.g., "... to the nearest 0.XX mm").

****We have added this information to the text.**

Line 153: Delete comma, to read "information through"

****Changed as suggested.**

Lines 224, 227, 264: Change "tibiofibular" (an adjective, as in "tibiofibular length") to "tibiofibula" (a noun). The term is spelled correctly on Line 281 and in Table 2.

****Changed as suggested.**

****We thank the reviewers for their very helpful comments.**

Journal Name: *Proceedings of the Royal Society B*

Journal Code: RSPB

Print ISSN: 0962-8452

Online ISSN: 1471-2954

Journal Admin Email: proceedingsb@royalsociety.org

MS Reference Number: RSPB-2020-1964

Article Status: SUBMITTED

MS Dryad ID: RSPB-2020-1964

MS Title: *The accelerating anuran: evolution of locomotor performance in cane toads (*Rhinella marina*, Bufonidae) at an invasion front*

MS Authors: Shine, Richard; Hudson, Cameron; Vidal-García, Marta; Murray, Trevor

Contact Author: Richard Shine

Contact Author Email: rick.shine@sydney.edu.au

Contact Author Address 1: Heydon-Laurence Building (A08)

Contact Author Address 2: Science Road

Contact Author Address 3:

Contact Author City: Sydney

Contact Author State: New South Wales

Contact Author Country: Australia

Contact Author ZIP/Postal Code: 2008

Keywords: *adaptation, Bufo marinus, jumping kinematics, dispersal, invasive species*

Abstract: *As is common in biological invasions, the rate at which cane toads (*Rhinella marina*) have spread across tropical Australia has accelerated through time. Individuals at the invasion-front travel further than range-core conspecifics, and exhibit distinctive morphologies that may facilitate rapid dispersal. However, the links between these morphological changes and locomotor performance have not been clearly documented. We used raceway trials and high-speed videography to document locomotor traits (e.g. hop*

distances, heights, velocities, and angles of take-off and landing) of toads from range-core and invasion-front populations. Locomotor performance varied geographically, and this variation in performance was linked to morphological features that have evolved during the toads' Australian invasion. Geographic variation in morphology and locomotor ability was evident not only in wild-caught animals, but also in individuals that had been raised under standardized conditions in captivity. Our data thus support the hypothesis that the cane toad's invasion across Australia has generated rapid evolutionary shifts in dispersal-relevant performance traits, and that these differences in performance are linked to concurrent shifts in morphological traits.

EndDryadContent